# The Role of Mineral and Trace Element Supplementation in Exercise and Athletic Performance: A Systematic Review

**DOI:** 10.3390/nu11030696

**Published:** 2019-03-24

**Authors:** Shane Michael Heffernan, Katy Horner, Giuseppe De Vito, Gillian Eileen Conway

**Affiliations:** 1School of Public Health, Physiotherapy and Sports Science, University College Dublin, D04 V1W8 Dublin 4, Ireland; katy.horner@ucd.ie (K.H.); giuseppedevito@ucd.ie (G.D.V.); 2School of Food Science and Environmental Health, Dublin Institute of Technology, Dublin 8, Ireland; Gillian.Conway@nibrt.ie

**Keywords:** ergogenic aids, nutritional supplements, physical performance, exercise and sport nutrition, muscle function

## Abstract

Minerals and trace elements (MTEs) are micronutrients involved in hundreds of biological processes. Deficiency in MTEs can negatively affect athletic performance. Approximately 50% of athletes have reported consuming some form of micronutrient supplement; however, there is limited data confirming their efficacy for improving performance. The aim of this study was to systematically review the role of MTEs in exercise and athletic performance. Six electronic databases and grey literature sources (MEDLINE; EMBASE; CINAHL and SportDISCUS; Web of Science and clinicaltrials.gov) were searched, in accordance with PRISMA guidelines. Results: 17,433 articles were identified and 130 experiments from 128 studies were included. Retrieved articles included Iron (*n* = 29), Calcium (*n* = 11), Magnesium, (*n* = 22), Phosphate (*n* = 17), Zinc (*n* = 9), Sodium (*n* = 15), Boron (*n* = 4), Selenium (*n* = 5), Chromium (*n* = 12) and multi-mineral articles (*n* = 5). No relevant articles were identified for Copper, Manganese, Iodine, Nickel, Fluoride or Cobalt. Only Iron and Magnesium included articles of sufficient quality to be assigned as ‘strong’. Currently, there is little evidence to support the use of MTE supplementation to improve physiological markers of athletic performance, with the possible exception of Iron (in particular, biological situations) and Magnesium as these currently have the strongest quality evidence. Regardless, some MTEs may possess the potential to improve athletic performance, but more high quality research is required before support for these MTEs can be given. PROSPERO preregistered (CRD42018090502).

## 1. Introduction

Minerals and trace elements (MTEs) are inorganic micronutrients found in a variety of plant and animal foods [1,2,3]. Inadequate MTE intake has been linked to a number of health conditions, such as diabetes, cardiovascular and kidney disease, aging and fracture risk [4,5,6,7,8,9,10,11,12,13]. These micronutrients are involved in hundreds of biological processes relevant to exercise and athletic performance, such as energy storage/utilization, protein metabolism, inflammation, oxygen transport, cardiac rhythms, bone metabolism and immune function [14,15,16,17,18]. However, despite the biological importance of MTEs, population data suggests that current RDAs are not being achieved, with Selenium, Magnesium, Calcium, Iron and Zinc of particular concern (60%, 50%, 51%, 30% and 17% reported deficiencies, respectively; [19,20,21]). While not adhering to RDAs does not strictly result in biological deficiencies, certain dietary and lifestyle choices may introduce additional challenges to RDA adherence and lead to deficiencies. The Western-Type diet (high in animal protein, saturated fats and refined carbohydrates) is the most adopted diet in first-world adult populations and shows deficiencies in Phosphorus (supplemented as Phosphate) and Magnesium [22]. The Atkins for Life diet, South Beach Diet, and DASH diet result in Chromium, Iodine and Molybdenum deficiencies [23]; Eat to Live-Vegan, Aggressive Weight Loss diet in Calcium, Selenium and Zinc; Fast Metabolism Diet in Calcium, Magnesium and Potassium; Eat, Drink and Be Healthy diets in Calcium and Potassium [24]; and a strict Vegan diet shows deficiencies in Iodine and Selenium [25]. However, a Mediterranean-style diet has been suggested to mitigate some of these deficiencies and may be superior to other diets for micronutrient intake [26].

The exposure of many exercisers and athletes to commercially available diets, via the wider and social media, can lead to the adoption of food choice in-line with these diets [27] and result in associated inadequate MTE intakes and deficiencies [28,29,30,31,32]. For example, some athletes (*n* = 25 male Polish) may be up to 60% deficient in the dietary intake of particular micronutrients [Magnesium (Mg); 33]. Similar to the available population data (for example, 15% deficiency of Mg; [22,23,24,25]), there are variabilities in the level of micronutrient deficiency in the diet, depending on the geographical location [31,33,34,35]. In an Australian population of elite female athletes (*n* = 72), Calcium (22%), Iron (19%) and Magnesium (15%) intakes were identified as deficient when assessed by a food frequency questionnaire [34]. Conversely, in a large Dutch population of sub-elite athletes (*n* = 553, female *n* = 226), the only deficiencies identified were Selenium (11%) in the whole group and Iron in females (38%; [35]), assessed by 24 h recall. These may be due to the mineral constituents of the dietary choices within each population [1,2,36] and the soil environments in different geographical locations [37].

There is also considerable potential for error in validity and reliability at all stages of dietary intake assessment, regardless of the method used [38]. Biomarkers can provide an objective assessment of nutrient status. However, among other limitations, few nutrients have reference ranges for well-trained athletes [38]. In situations of high metabolic demand, such as exercise or athletic training, inadequate circulating and cellular MTEs may impair optimal physiological performance [14,30] and may require supplementation [17]. The exact impact of these deficiencies or supplementation on athletic performance remains generally unclear [39]. However, there may be ergogenic properties of MTEs in achieving or possibly surpassing the RDA, that are specifically designed for the general population health. While there is currently no consensus as to the efficacy of MTE supplementation for exercise and any physiological measure of athletic performance, recent studies show that ~50% of athletes consume some form of micronutrient supplement and between 5–27% are MTEs (Iron, Calcium, Zinc, Selenium or Chromium; [31,40]). Synthesis of some MTE research, in the context of athletic performance, can be found throughout the literature [41,42,43,44,45,46,47,48]; however, there are some MTEs and more recent studies that have yet to be systematically reviewed. Collating the current evidence on the efficacy of MTE supplementation for athletic performance in a single article is warranted and may provide a useful tool for improving the knowledge base of athletes and sport/exercise practitioners; specifically, the effects of MTEs on those phenotypes that benefit general markers of performance, for example, a lower body power output, maximal endurance capacity, maximal and relative muscle mass and strength, and fatigue recovery capacity.

Therefore, the present aim was to systematically review the literature and critically synthesise the available evidence on MTE supplementation for enhancing exercise and physiological aspects of athletic performance. In addition, this review aimed to make recommendations about the efficacy of MTE supplementation for optimising athletic performance, based on the quality of the retrieved research.

## 2. Methods

The review was conducted in accordance with the Preferred Reporting Items for Systematic Review and Meta-Analysis (PRISMA, checklist in Appendix B) statement [49] and preregistered with PROSPERO (CRD42018090502). 

### 2.1. Search Strategy

A systematic search of six electronic databases and grey literature sources (MEDLINE; EMBASE; CINAHL and SportDISCUS; Web of Science and clinicaltrials.gov) was performed using predefined search terms deduced from the eligibility criteria and PICO guidelines [50] between January and April 2018. The reference lists of identified reviews and included articles were hand searched for potentially relevant articles. Where appropriate, the search was conducted using Medical Subject Headings and Boolean operators of keywords relating to population (athlete etc.), intervention (Calcium etc.), and outcome (athletic performance etc.; Appendix C).

### 2.2. Study Selection and Data Extraction

Following the initial data search, a complete search of titles and abstracts was performed by two independent reviewers (SH and GC). Titles and abstracts were screened for eligibility according to predefined inclusion and exclusion criteria. In the case of an inclusion discrepancy, the two independent reviewers discussed the merits of selection. If a consensus could not be reached, a third reviewer (KH) was consulted to resolve the issue and an agreement was achieved. Retrieved articles in fulfilment of the inclusion criteria were accessed, in full, and critically appraised, and the article data was extracted using a customised form (content of data extraction was decided by two reviewers) and characterised into methodological (risk of bias etc.), participant characteristics (age etc.) and study characteristics (sample size etc.). In any cases where more than one distinct experiment was performed in a single article, the experimental data for each experiment was extracted and assessed separately (for example, in Shae et al. [51]).

### 2.3. Eligibility Criteria

Studies of healthy adult and athletic populations (+17 years), English language, both sexes and supplement studies were included. Reports on animals, cells, children, diseased populations, dietary intake only and psychological phenotypes were excluded. Diseased populations where mineral deficiency was part of the diagnosis/prognosis were excluded; however, studies including individuals with deficiencies that were otherwise healthy were included. Interventional and control trials were the target designs for the present review to ensure the capture of all relevant articles. The following article types were excluded: editorials, systematic reviews, letters to the editor, commentaries, duplicated publications and articles that combine minerals with other molecules such as the amino acid aspartic acid combined with Zinc and Magnesium in ZMA [52]. In additionally retrieved (not identified through database searches) articles, the references list(s) were scanned for appropriate original articles.

### 2.4. Quality Assessment and Risk of Bias Tool

The Effective Public Health Practice Project Quality Assessment Tool (EHPP; [53]) was used to assess study quality and risk of bias independently by two authors (SH and GC). On the occasion of a discrepancy in the global quality, a third independent reviewer (KH) assessed the article(s) and a consensus was achieved. To ensure selection consistency and quality assurance, a random sub-sample of retrieved studies were cross-checked, post-hoc and independently, by KH (blinded to the original review decisions). For studies describing athletes as ‘well/highly/trained/elite’, selection bias was graded as “Somewhat Likely” (this decision was made in consultation with the EPHPP licence holders at McMaster University).

## 3. Results

A total of 17,459 articles were identified, and after removing duplicates, 14,144 articles remained. Of these, 13,816 were excluded after title and abstract screening. The remaining 328 articles were screened in full text and 128 studies met the eligibility criteria (Figure 1).

### 3.1. Study Characteristics

The 128 eligible studies consisted of 3643 participants (1387 Females), aged between 17–75 years, and included 24 studies of elite athletes (Appendix A). The eligible studies related to Iron (Fe; *n* = 29), Calcium (Ca; *n* = 11), Magnesium, (Mg; *n* = 22), Phosphate (P; *n* = 17), Zinc (Zn; *n* = 9), Sodium (Na; *n* = 15), Boron (Br; *n* = 5), Selenium (Se; *n* = 5), Chromium (Cr; *n* = 12) and Multi-mineral articles (*n* = 5). No relevant research articles were identified for Copper, Manganese, Iodine, Nickel, Fluoride or Cobalt.

### 3.2. Study Quality

Using the quality assessment tool, eight articles were identified as strong, 95 as moderate and 25 as weak. Of these, only Fe and Mg included articles of sufficient quality to be classified as strong (Fe = 6, Mg = 2; Figure 2). Overall, the majority of retrieved articles were assigned a moderate quality (77%, Figure 2 and Appendix A).

## 4. Discussion

This systematic review aimed to synthesise the evidence relating to the effects of MTE supplementation on athletic performance and related physiological phenotypes in the adult population. Quality of evidence investigating MTEs and athletic performance is lacking, with only eight articles classified as strong. Nonetheless, there is limited but growing evidence for potential benefits of some MTEs in relation to athletic performance (Fe and Mg), although the majority of research quality remains moderate-weak (Figure 2). One article was identified that presented evidence for a possible benefit of a particular combination of minerals on athletic performance-related phenotypes (see Section 4.10). Furthermore, the present review retrieved recent, although still limited, evidence for a ‘natural’ mineral-rich compound *Lithothamnion* and its potential for athletic performance-related haematological phenotypes, although currently no evidence for functional performance.

### 4.1. Iron

Twenty nine articles fulfilled the inclusion criteria for iron (Fe), including 946 participants (females, *n* = 776). These consisted of 19 randomised control trials (RCT’s), 10 of which referred to elite athletes. Iron is the most studied mineral in exercise and athletic performance, with the best quality research (Figure 2). Non-anaemic Fe deficiency (serum ferritin <20.0 µg·L^−1^, Hb >115 µg·L^−1^) and anaemia (serum ferritin <12.0 µg·L^−1^, Hb <115 µg·L^−1^) are common at all levels of athletic performance and are thought to affect physiological capacity. Fe deficiency with and without anaemia has been repeatedly shown to be attenuated following both oral and intravenous (IV) Fe supplementation in a variety of sports [54,55,56,57,58,59]. However, the potential benefits of Fe supplementation on physiological performance may be dependent of baseline ferritin level, Fe dose and route of administration.

Baseline iron status or ferritin level is a major factor that could impact the efficacy of Fe supplementation on Fe status and performance-related outcomes. For example, in a group of elite endurance athletes (*n* = 178, 80 females) divided by baseline ferritin levels prior to training at moderate altitude, those with high ferritin levels (>100 µg·L^−1^) were given no supplement, mid-ferritin levels (~76 µg·L^−1^) were supplemented with 105 mg Fe and low ferritin levels (~25 µg·L^−1^) were supplemented with 210 mg for two to four weeks [60]. Haemoglobin mass (HBmass) increased in those with low and mid-baseline ferritin levels supplemented with Fe, but there was no change in non-supplemented athletes. In addition, follow up ferritin levels increased by 37% in the group with the lowest baseline levels of ferritin, whereas in the other groups, ferritin decreased and total erythrocyte Fe uptake increased as a result of moderate altitude [60]. In female distance runners and triathletes with low baseline ferritin levels, high dose Fe (350 mg of ferrous gluconate) over eight weeks following recovery from two weeks of intensive training similarly increased serum ferritin levels, but had no effects on serum Iron, haematocrit or markers of immune function (natural killer cells), compared to calcium carbonate [61]. Hinton and Sinclair [62] also showed that daily 30 mg of oral Fe in 20 NAID males and females (ferritin <16 µg·L^−1^) for six weeks did not alter VO_2peak_, but improved serum ferritin, energetic efficiency during submaximal exercise and attenuated the decline in ventilatory threshold that was observed with the placebo. Elsewhere, in active females with low ferritin, eight weeks of Fe supplementation (100 mg·d^−1^) increased serum ferritin, Hb, VO_2peak_ and reduced blood lactate following submaximal exercise [63]. Others have shown that in male runners with relatively high ferritin (~61 µg·L^−1^), assessed throughout a 20-day 500 km road race with low-dose Fe supplementation (36 mg·day^−1^), serum Iron and ferritin did not change significantly [64]. Furthermore, in non-Fe-deplete (ferritin >70 µg·L^−1^) elite male boxers, 1335 mg of ferrous-glycine-sulphate (equivalent to 200 mg elementary Fe) had no effect on physiological parameters (VO_2peak_ and other ventilatory markers) measured at moderate altitude (18,000 m) over 18 days [65]. Similarly, in non-deplete (Ferritin >100 µg·L^−1^) exercising females, 12 weeks of low dose oral Fe (50 mg·day^−1^) had no effect on VO_2peak_ [66].

It is important to acknowledge, however, that not all studies show improvements in iron status or endurance performance-related outcomes, such as VO_2peak_ or lactate levels, in those with low ferritin levels. For example, in female athletes with low (<26 µg·L^−1^) ferritin, no improvement in VO_2peak_, lactate concentrations or time to exhaustion has also been shown following eight weeks of supplementation (100 mg·day^−1^) compared to the placebo, despite improvements in iron status [67,68]. In an older study using extremely low Fe over eight weeks (9 and 18 mg·day^−1^ - the RDA of the day; [69]) in females with a wide range of baseline ferritin levels, Fe had no effect on serum ferritin, Iron, Hb or haematocrit. Elsewhere, in anaemic and non-anaemic females with a wide range of baseline ferritin levels, somewhat of a high dose (160 mg·day^−1^ ferrous sulphate) during a 42 day period of intense physical training (5–6 h per day, 6 days per week) maintained ferritin and resulted in a significant increase in VO_2peak_ at three weeks, but not at six weeks, compared to the placebo [70]. However, it was not possible to separate the findings by baseline iron status, limiting the interpretation in this regard. Collectively, although some inconsistency in findings are evident, the majority of studies that show benefits of Fe supplementation on measures of iron status and outcomes related to endurance performance are in individuals with low baseline ferritin levels, whereas there is little evidence for beneficial effects when Iron status is not compromised.

In the context of “real-world” athletic performance, Fe has been shown to have positive effects on measures of functional performance. An improvement in a 15 km time trial was demonstrated following four weeks of high intensity endurance training (75–85% max heart rate) with six weeks of supplementation (100 mg·day^−1^) and was accompanied by increases in serum ferritin and Hb in NAID females [71]. In a large study (*n* = 171) of female military recruits undergoing eight weeks of basic combat training, the sample was divided into normal Iron status, NAID and Fe-deficient-anaemia, with each group receiving either a low dose of 100 mg of ferrous sulphate or placebo [72]. The supplemented Fe-deficient anaemia group completed a two mile time trial 110 seconds faster and improved indicators of mood, particularly ‘vigour’. Elsewhere, amateur female rowers (Ferritin <29 µg·L^−1^) undergoing six weeks of specific training with 100 mg·day^−1^ Fe showed a slower rate of lactate accumulation at the 1 km and 2 km stages of a 4 km time trial and recovered quicker (~7%) at 5 min post exercise [54]. Furthermore, in those with the lowest baseline Fe (Ferritin <20 µg·L^−1^), Fe improved 4 km time trial energy efficiency (kcal) more than the placebo, although this was not observed in those with a higher Fe status [54]. Others have shown that in endurance trained females with very low baseline ferritin levels (14–18 ng·dL^−1^), taking 60 mg of Fe·day^−1^ over eight weeks of intense training 3000 m time (~10 m·min^−1^) improved along with running velocity to lactate threshold and onset of blood lactate accumulation improved-but there was no change in VO_2peak_ compared to the placebo [73]. These findings collectively show the benefits of Fe supplementation on some measures of functional performance in individuals with low baseline ferritin levels.

Other performance-related outcomes, including fatigue resistance and strength, have also been shown to improve following Fe supplementation in females with low ferritin levels, and in studies of females with varying Iron status. Brutsaert et al. [74] investigated the effect of six weeks of oral Fe (20 mg·day^−1^ elemental Fe) on maximal voluntary contractions (MVC) following a quadriceps muscle fatiguing protocol in NAID females (ferritin <15 µg·L^−1^; *n* = 20) over six weeks. Fe increased post-fatiguing MVC by 15% and showed a significant improvement in fatigue resistance in quadriceps MVC, compared to the placebo. In elite volleyball players who were NAID or had adequate iron stores that were supplemented with a high dose (105 mg·day^−1^ elemental Fe) for 11 weeks during the competitive season, markers of dynamic strength, the clean and jerk, power clean and total mean strength performance (ranging between ~10–40%) were enhanced and Iron loss was prevented compared to controls [75].

Along with dose, the route of Fe administration may also influence the efficacy of supplementation in individuals of varying Iron status. Although used since the 1970’s [76], recent advances have shown the rise in the use of parenteral Fe preparations, which appear superior to oral supplementation in enhancing measures of athletic performance [77], but not ubiquitously [42]. Six weeks of IV was superior to high-dose oral Fe for improving VO_2max_ (by 2.5%) and max running time (3.7%) in non-anaemic endurance runners (ferritin ≤65 µg·L^−1^ and [Hb] >12 g·dL^−1^) [77]. Furthermore, when subdivided into low (ferritin <35 µg·L^−1^ and transferrin saturation <20%, or ferritin <15 µg·L^−1^) or everyone else (classified as ‘suboptimal’), HBmass (~4–5%), VO_2max_ (2%) and max running time (6%) increased more in the low IV group compared to the oral low or suboptimal IV group [77]. Others have shown an improvement in serum ferritin but no performance effect of IV Fe compared to the placebo [78]. However, this may be due to methodological limitations, including that follow up testing was performed ~10 days after the last supplementation period and the testing was not standardised to facilitate the athletes’ disciplines. In highly trained distance runners (6 male, 8 female) without clinical Fe deficiency (~60 km·week-training^−1^; ferritin 30–100 µg·L^−1^), three Fe injections over four weeks had no effect on 3000 running performance, but improved perceived fatigue and mood compared to saline injections [79]. Therefore, the current evidence on whether Fe IV infusion or injections are superior to oral supplementation is limited.

In conclusion, there appears to be a range of factors, including baseline Iron status, dose and route of administration, that may influence the efficacy of Fe supplementation on Iron status and performance. It is generally accepted that Fe increases HBmass, leading to greater oxygen delivery; however, there may be other mechanisms not related to erythropoiesis and oxygen transport at play (i.e. no change in HBmass in non-Fe deficient athletes [77,79]) that, over longer trial periods this could lead to performance enhancement. The evident general trend, as is also evident in recent systematic reviews, is that Fe supplementation may benefit Iron status and athletic performance in individuals with a compromised Iron status [42,43]. In NAID females, oral elemental Fe supplementation between ~15–60 mg·day^−1^ or 100 mg·day^−1^ of ferrous sulfate over six to eight weeks may be adequate to elect performance adaptations in 3000 m running time, running velocity to lactate threshold, onset of blood lactate accumulation, quadriceps MVC fatigue resistance, post-fatiguing MVC, 4 km rowing time trial energy efficiency (kcal), preventing exercise-induced Fe loss, Hb, VO_2peak_, improving 15 km cycling and two mile running time trial, quicker recovery post exercise and indicators of mood. Approximately 100 mg·day^−1^ elemental Fe over 11 weeks has been shown to result in adaptations in markers of dynamic and absolute strength. Intravenous Fe administration may be beneficial to improving running time and VO_2max_ performance; however, the current evidence on whether Fe IV infusion or injections are superior to oral supplementation is limited and the administration (i.e. via a medical professional) makes it difficult to recommend such approaches at this time. Future studies should focus on recruiting larger samples, including elite athletes, tracking longer term supplementation and considering alternative mechanisms to the potential changes in HBmass. As the majority of individuals supplementing with Fe are not elite athletes, investigations of Fe-associated training adaptations in non-elite individuals are also currently limited and warrant attention.

### 4.2. Calcium

Eleven articles fulfilled the inclusion criteria for Calcium (Ca), including 311 participants (females, n = 58). These consisted of 19 RCT’s, one of which referred to elite athletes. Exercise is known to induce modest Ca loss following non-weight bearing steady-state activities; however, Ca supplementation might mitigate this loss. Ca loss can also lead to hormonal changes [80] that might impair muscle function; however, limited quality data exists for direct markers of functional performance.

An investigation of Ca supplementation on Ca homeostasis during exercise in healthy active premenopausal women demonstrated that Ca supplementation (800 mg above the controlled allowance for placebo) effectively attenuated exercise-induced Ca loss [81]. In fact, Ca supplementation altered the usual Ca loss associated with exercise into positive retention. These data are important because cellular Ca is vital for skeletal muscle function, particularly the calcium-dependent troponin complex, and parathyroid hormone (PTH) homeostasis [82,83]. The precise impact of circulating Ca on muscle function and exercise/athletic performance, however, remains unclear. For example, additional Ca (35 mg·kg·day^−1^; standard intake ~1800 mg·day^−1^) did not affect Ca balance or VO_2peak_ performance during normal endurance training (67 km·week^−1^) or when trained endurance athletes were restricted to 7 km·day^−1^ for 12 months [84]. However, with restricted activity, iPTH (ionised PTH) decreased to a greater extent in the Ca supplemented group compared to the non-supplemented group [84]. In a very similar study design, Zorbras et al. [85] also showed that when athletes considered to be Ca deficient at baseline (~1500 mg·day^−1^ dietary intake) restricted training (0.7 km·day^−1^) and consumed 55 mg·kg·day^−1^ Ca over 12 months, circulating PTH was reduced.

Continual IV infusion of Ca (156 mg), with the intention of maintaining consistent blood Ca during exercise [86], has also been shown to attenuate the usual exercise-induced increase in PTH by 65%, compared with a saline infusion. In agreement, elite athletes consuming either high (972 mg·L^−1^) or low (18 mg·L^−1^) Ca water during exercise found similar effects for PTH (at peak concentrations, Low Ca = 36 pg·mL versus High Ca = 65 pg·mL; [87]). This is potentially important for exercise and athletic performance as high PTH levels have been linked to significant muscle impairments [88,89], PTH replacement in hypothyroid patients can decrease maximal muscle strength by ~30% and can impair neuromuscular motor unit action potentials [90].

Another route to ensure adequate or supplementary Ca intake is through food control and whole food supplementation. Haakonssen et al. [91] provided female road cyclists with a meal consisting of either 46 mg (control group) or 1352 mg (Calcium group) of dairy Ca 90 mins prior to a 90-min cycle trial. Blood samples were taken pre-trial; immediately pre-exercise and at 40 min, 100 min and 190 min post-exercise. Serum iCa decreased with the onset of exercise, but was higher in the Ca supplemented group (by ~0.040 mmol·L) immediately prior to exercise and remained higher post-exercise and at 40min post-exercise [91]. Similarly, iPTH was persistently lower in the Ca supplemented group compared to controls prior to, post-exercise and during recovery - to the largest extent immediately post-exercise (1.55 times lower) and was independent of Ca loss through sweat.

Cinar et al. [92,93,94,95,96] investigated the influence of Ca (~37 mg·kg·day^−1^) and exhaustive exercise (90 min·day^−1^, 5 days·week^−1^) on a number of exercise-related blood markers, in a series of studies in the same sample of amateur athletes (*n* = 30). The authors concluded that circulating Ca, Potassium (K), Copper (Cu), Testosterone (T), Glucose, Leukocyte and Erythrocyte levels were altered following exhaustive exercise and occurred to a greater extent with exercise plus Ca supplementation [92,93,94,95,96]. However, these conclusions are misleading, as the presented results suggest that it was more likely that exhaustive exercise that influenced these changes, rather than Ca supplementation. For example, the exercise condition (no supplementation) increased circulating Cu by ~0.34 mg·dL and circulating T by ~4 pg·mL; whereas Ca supplementation plus exercise increased Cu by ~0.85 mg·dL and T by ~3 pg·mL and Ca alone resulted in the opposite—a decrease in Cu and T (data derived from [92,93]). This trend continued for Leukocyte and Erythrocyte levels, showing that Ca supplementation likely had no effect on exercise-related blood biomarkers; rather, exhaustive exercise was the driver for the observerations. Furthermore, the authors found no effect of Ca supplementation and/or exhaustive exercise on plasma adrenocorticotropic hormone (ACTH) and cortisol levels [95]. This data is curious as it has been known for some time that both biomarkers increase following exercise, particularly exhaustive exercise [97].

Although Ca is vitally important for muscle and cardiovascular function [98,99], there is currently no evidence that Ca supplementation has any direct effect on athletic performance (currently, only aerobic capacity has been investigated). Nonetheless, calcium supplementation at oral doses between 800 (over 8 days) to 1352 mg (single meal prior to exercise), or IV infusion at 156 mg (prior to and during exercise), may attenuate post-exercise reductions in serum iCa and Ca loss, with lower doses appear to have no effect. Supplementary Ca may also reduce the exercise-induced increase in iPTH. As mentioned above, the impact on PTH could potentially have implications for muscle strength [88,89]. However, there is currently no direct evidence to support the hypothesis that Ca supplementation may enhance muscle physiological capacity through the actions of PTH. Future research should investigate this using detailed measures such as hormonal changes, muscle fibre characteristics, muscle-derived biochemical effects, action potentials, and functional and specific strength measures.

### 4.3. Magnesium

Twenty two articles fulfilled the inclusion criteria for magnesium (Mg), including 663 participants (females, *n* = 72). These consisted of ten RCT’s, three of which referred to elite athletes. Evidence is growing that Mg may be an important element to maintain muscle mass, power and markers of systemic inflammation [100,101], although the effects of supplementation on these parameters remains ambiguous [102]. There appears to be some inconsistencies in the literature that are likely to be a result of dosing, baseline mineral status/intake, exercise intensity and population. For example, low dose Mg supplementation may be simply elevating Mg to the required physiological levels, rather than being sufficient to have an ergogenic affect. This is reflected in the general trend for higher doses to have more positive results [103,104] than lower doses [105,106,107,108].

Exercise is known to effect Mg metabolism and there is an alternating response, depending on exercise intensity [107,109,110]. In elite handball athletes, greater time spent exercising at low-to-moderate intensity was associated with higher plasma Mg levels, whereas a greater time spent training at >80% residual heart rate was associated with lower Mg levels (*r* = 0.38, *p* < 0.01; [109]). Intense training over four weeks similarly resulted in lower Mg levels in elite volleyball players, including in those supplementing with Mg [107]. However, others have shown that supplementing with 400 mg·day^−1^, in addition to 217 mg per 1000 kcal of dietary Mg, maintained Mg status throughout an elite basketball competitive season, except during the most competitive portion of the season [110]. These data imply that with adequate Mg availability and under conditions of low physiological demand, Mg is released into the circulation, likely from muscle, but not necessarily used. Whereas, when stores are inadequate to supply the demand in conditions of physiological stress, circulating Mg levels decrease, highlighting the potentially important role of muscle in Mg metabolism. A responder/non-responder paradigm has also been suggested to play a role in Mg and exercise metabolism, and requires further explanation [111].

The exact relationship between blood and muscle Mg is unclear and also requires further investigation. When well-trained endurance athletes were restricted from running (13.9 km·day^−1^ to 4.7 km·day^−1^) for 12 months, they entered a negative Mg balance (Mg intake lower than faecal and urinary losses), compared to a positive balance in athletes where exercise was non-restricted. This negative balance resulted in increased serum Mg, irrespective of low dose Mg supplementation (0.5 mg kg day^−1^; [112]). Interestingly, the negative balance was greater when reduced activity was combined with Mg supplementation and the greatest positive balance was evident in the athletes who combined Mg supplementation with their usual training. These data suggest that when Mg is supplemented during exercise restriction, significant amounts are excreted, likely because lower amounts of Mg are being utilised. This highlights the role of muscle in Mg metabolism.

Two older studies have investigated muscle-derived Mg in the context of supplementation and post-exercise requirement [113,114], with no exercise-associated change in muscle-derived Mg content identified. In one study, Mg supplementation (365 mg day^−1^) had no effect on muscle or serum Mg concentrations or on 42 km marathon running performance compared to a placebo group [113]. However, the post-exercise muscle biopsies were take 48 h after the marathon and by this time, Mg concentrations are likely to have returned to resting levels. Weller et al. [114] investigated the effect of Mg supplementation (500 mg day^−1^) over three weeks on muscle, serum, leukocyte and other blood-derived cell Mg in a group of athletes with low-normal serum Mg levels. There was no effect on muscle, serum or blood Mg concentrations and no difference in aerobic capacity, neuromuscular function or exercise haematological parameters between Mg and placebo groups. However, there was a weak correlation between muscle Mg (measured by NMR) and total Mg in mononuclear leukocytes, and an inverse correlation between muscle and red cell Mg. It appears that serum Mg does not necessary reflect muscle Mg (although serum is often measured), however exercise was not well-recorded and muscle Mg was not measured following exercise. These data are interesting but owing to the lack of rigorous methodology, the effect of exercise with Mg supplementation on muscle-derived Mg is currently unknown and warrants future investigation.

The effects of Mg supplementation on functional performance outcomes and related measures in general appear inconsistent. For example, there is currently no evidence to support Mg supplementation enhancing endurance capacity [100,106,107,112,114], despite a logical biological potential [115]. However, positive effects have been shown in some other outcome measures. In elite volleyball athletes, Mg supplementation (350 mg·day^−1^) over four weeks improved (~6%) countermovement jump, compared to the placebo, although there was no difference in neuromuscular capacity (Isokinetic dynamometry) [107]. Kass et al. found improvements in blood pressure at rest and during recovery following Mg supplementation over two weeks (>300 mg·day^−1^), but no effect on isometric bench press or endurance performance indicators [105]. However, in a follow up study, Kass and Poeira [116] investigated the effect of acute (300 mg·day^−1^ for one week) and chronic (4 weeks) Mg supplementation compared to a placebo on exercise and recovery from resistance exercise in 13 recreational endurance athletes. These athletes were already consuming ~370 mg·day^−1^ of dietary Mg and the results showed that the effects may depend on duration of supplementation. A 40 km cycling time-trial was carried out to elicit physiological stress, deemed typical training, where blood pressure and 1 repetition maximum (1RM) bench press (following the time-trial) were assessed at baseline and over two consecutive days following the supplementation period. Acute Mg increased 1RM by 17% compared to baseline (prior to commencing supplementation), whereas in the chronic (over 4 weeks) Mg group, there was no change in 1RM bench press. In addition, acute Mg showed no decline in force during repetitions to fatigue the next day following the time-trial, whereas in the chronic Mg group, there was a 32% performance decrement. Blood pressure also showed a greater and more consistent reduction in the acute versus chronic Mg groups [116]. However, changes in neuromuscular strength may be influenced by a higher dose and duration of Mg supplementation with training intensity. In previously untrained participants, eight mg·kg·day^−1^ Mg (achieving ~144% of the RDA compared to 70% in the control group) for seven weeks, together with lower body resistance exercise three times per week, improved absolute (~40 Nm) and relative (~0.9 Nm·kg of lean body mass) strength compared to a control group [103]. In contrast, in an older population (>65 years) undertaking a ‘mild fitness program’, twelve weeks of 300 mg·day^−1^ (*n* = 62) had no effect on neuromuscular or handgrip strength compared to a control group (no placebo or intervention, *n* = 77; [117]). However, other markers of physical function improved (Short Physical Performance Battery; chair stand, 4m walking test) following Mg supplementation and the improvements were more pronounced in those with a lower Mg intake. Combined, these data suggest (although relatively weakly with the current evidence) that in younger populations, there may be a capacity for high-dose Mg to elicit improvements in functional markers of athletic performance and with longer trial periods, neuromuscular strength. Whereas in older populations, moderate dose Mg can improve functional markers of health and standard markers of physical function.

There appears to be some evidence that Mg can also positively influence exercise-induced haematological changes. Four weeks of 500 mg·day^−1^ of Mg in an amateur rugby union ameliorated the exercise-induced increase in IL-6 (a marker of systemic inflammation), reduced white blood cell count, neutrophils percentage, post-game cortisol, increased adrenocorticotropic hormone and lymphocytes percentage, compared to controls [104]. These trends were evident at various time-points throughout six days of recovery following a game. Interestingly, a significant reduction in cortisol was observed in the Mg supplemented group on the day prior to and the morning of the game, but not on the day after the game [104]. The same researchers, using a similar design, showed that Mg mitigated exercise-induced DNA damage in the presence of H_2_O_2_, but not without, following the same amateur rugby game [118]. Recent evidence has shown the importance of some of these molecules in exercise adaptation [119,120,121], however the implications of these changes for athletic performance require further investigation.

It should also be noted that Cinar et al. presented findings on the effects of Mg (10 mg·kg·day^−1^ over 4 weeks) and exercise training on haematological measures of immune function, and hormonal, insulin and glucose status at rest and following an exhaustive exercise protocol [122,123,124,125,126]. The authors concluded that Mg supplementation in “sportsmen” elicited higher circulating Mg and Zn [123], blood cell count were reduced; erythrocytes were increased [123], and leukocytes, erythrocytes, ACTH, cortisol, glucose and testosterone were increased both at rest and following exercise to exhaustion [122,124,125,126]. However, as noted with other papers by these authors, these conclusions should be taken with caution as many of the biomarker changes were likely the result of the training conditions rather than Mg supplementation.

In conclusion, the current evidence suggests that 300–500 mg·day^−1^ for short-term supplementation (~1–4 weeks) can have a positive influence on functional dynamic measures of muscle performance (CMJ, 1RM and fatigue resistance) and exercised-induced inflammation, DNA damage, cortisol and immunological blood markers, but no effect on isokinetic performance [107,116,118]. Whereas, longer supplementation trials (~7 weeks) can elicit training-induced adaptive responses in young untrained populations [103]—whether this would be reflected in well-trained athletic populations is yet to be established. Furthermore, Mg (300 mg·day^−1^ over 12 weeks) may improve markers of functional performance in older populations and may be a consideration to maintain functional capacity throughout aging, but may require an even longer treatment period [127]. Lastly, there is currently no evidence to support a benefit of Mg supplementation to improve endurance performance-related outcomes, despite a logical biological potential [115]. However, the literature is limited and further research is required. Mg appears to have some ergogenic potential, but much more evidence is needed in a variety of populations (untrained, elite athletes and elderly) and in response to both aerobic and resistance/dynamic power training and performance. Little is currently known about the direct effect of Mg ingestion on muscle in response to exercise and further investigation is needed to uncover the mechanisms of action for the above physiological responses.

### 4.4. Phosphate

Seventeen articles fulfilled the inclusion criteria for phosphate (P). These included 247 participants (females, *n* = 48), consisting of 13 RCT’s, two of which referred to elite athletes. Phosphate (commonly supplemented as sodium phosphate (SP); the supplementary form of phosphorus) has been shown to improve a range of parameters, including sprint time, cycling power output, VO_2Peak_, resting HR, biomarkers markers of metabolic demand and measures of cardiac function (echocardiographic).

Several studies have shown improvements in sprint time, total work and power output in a range of athletes. In recreational male (*n* = 11) and female (*n* = 12) team sport athletes, supplementation with 50 mg·kg·FFM^−1^ of SP over six consecutive days was superior for improving sprint performance compared to a placebo using magnitude-based inferences [128,129]. Participants performed a simulated team-game circuit with a 6 × 20 m repeated sprint set performed before, at half-time and at the end. In males, SP resulted in faster times for all sprints compared to the placebo, caffeine or a combination of SP plus caffeine (Cohen’s *d*′ = 0.5–0.8), and in females, both SP and combined SP plus caffeine improved repeated sprint ability compared to placebo [128,129]. The same researchers showed that in female team sport athletes (*n* = 13), SP was superior (Cohen’s *d*′ = 0.5–0.8) in improving total sprint times for all sets and overall, and the best sprint times were improved for all sets, with ~6% improvement after SP compared to the placebo and beetroot juice [130]. This group also demonstrated improvements in total work and power output in trained cyclists in a race simulation on days 1 and 4 following six days SP supplementation, compared to no change in the placebo [131]. Others have also shown improvements in power output by ~30 W and finishing time during a 1 km time trial in well-trained cyclists following six days SP (4 g day^−1^) supplementation, compared to a placebo (*n* = 7; [132]).

A range of outcome measures related to endurance performance, including VO_2peak_ and blood lactate, have also been shown to improve following SP supplementation. In competitive male cyclists, SP loading significantly improved VO_2peak_ and interestingly, a second loading phase separated by a washout period (15–35 days) resulted in even greater improvements, compared to a placebo (*n* = 12; [133]). An older study comparing two days of potassium phosphate (PP) loading (4 g·day^−1^) to placebo in highly-trained endurance runners (crossover *n* = 8) showed that PP may reduce the rate of perceived exertion (RPE) during the mid-stages of maximal treadmill running, although no changes in physiological parameters were identified. This may be due to the short loading phase [134]. Others have shown that SP loading (4 g) over three days improved VO_2peak_ in elite endurance athletes by ~9%, increased blood haematocrit levels, glucose and other markers of metabolic demand [135]. SP also enhanced echocardiographic measures of endurance performance by 5–12% (ejection fraction and fractional shortening). These data are an indicator of the possible mechanisms that SP may have on athletic performance (in competitive male endurance athletes, *n* = 6). Particularly as low and high P has been linked to impaired echocardiographic measures in critically diseased populations [136,137,138,139,140], this emphasises the importance of P in cardiac function. Surprisingly, no recent echocardiographic data related to SP and athletic performance was identified in the present review, but would be of interest for future investigations.

Recent evidence suggests that SP appears to maintain the exercise-associated benefits of supplementation when the loading phase (50 mg·kg-FFM·day^−1^ for 6 days) is followed by a lower dose (25 mg·kg-FFM·day^−1^) for a subsequent three weeks [141,142]. In elite off-road mountain cyclists, the loading phase increased VO_2peak_ (by 5.3%), and reduced resting HR (by 9.6%), max HR (by 2.7%), and HR at lactate threshold (1.7%), all of which were maintained following lower dosing, with no change in placebo [142]. Furthermore, maximal power output did not change due to the loading phase, but increased significantly after the maintenance phase, implying a possible delayed response to the peripheral tissues (as described with Zinc). However, to date, this has not been experimentally shown. This adaption could be a result of increased 2,3-diphosphoglycerate (2,3-DPG) concentrations as 2,3-DPG decreases the affinity of Hb for oxygen, thus resulting in the greater unloading of oxygen to the peripheral tissues [142] and increased 2,3-DPG after SP loading [142,143,144].

It is interesting to note that “moderately” trained individuals supplemented in a similar way (6-day loading) showed no effect on aerobic capacity or power output [145,146,147,148]. This suggests that SP may only improve aerobic performance when individuals are well-trained or elite athletes, indicating that the dietary ingestion of phosphorus may be adequate outside of extreme physiological requirements. This may be reflected in the relatively unchanged blood phosphate levels reported by some authors following supplementation in well-trained athletes [131,133,135]—although this has not been observed by all [134,142]. It is important to note, as suggested by Kreider et al. [135], that serum phosphate concentrations may not accurately assess the effects of phosphate loading on intracellular phosphate levels and oxidative metabolism. However, no investigation of muscle-derived P activity was retrieved in this review.

In conclusion, the current evidence indicates several ergogenic effects of SP supplementation with ~4 g·day^−1^ over three-to-six days on a range of performance-related outcomes, such as sprint time, cycling power output, VO_2Peak_, resting HR, biomarkers of metabolic demand and measures of cardiac function (echocardiographic). However, these ergogenic effects are limited to highly-trained individuals and may not aid recreational athletes when dietary phosphorus is adequate and outside of the requirements of highly-trained athletes. In addition, benefits in endurance performance-related outcomes can be further maintained with a lower dose of ~2–4 g·day^−1^ [141,142]. As there is some evidence for a delayed response to the adaptation of muscle power [142], this is an interesting area for future investigation, along with determining if this is a central or peripheral physiological adaptation and if it is directly related to changes in 2,3-DPG. Furthermore, Kreider et al. [135] presented some possible mechanistic data in relation to morphological changes to the myocardium in a small sample, but this requires replication with a larger sample in future investigations.

### 4.5. Zinc

Nine articles fulfilled the inclusion criteria for zinc (Zn). These included 229 participants (females, *n* = 50), consisting of four RCT’s, one of which referred to elite athletes. Few studies exist that have directly assessed the effect of Zn (without other molecules) on athletic performance phenotypes and the majority appear to assess physiological variables that indirectly contribute or augment exercises-induced reduction in immunity [149,150,151].

Nevertheless, six weeks of low-dose Zn (gluconate; 30 mg·day^−1^) have been shown to improve estimated VO_2peak_ (Bruce protocol), similar to that of HITT training (~4%; additional to team training) and when combined, Zn and HITT further improved estimated VO_2peak_ (by 9.33%) in female futsal athletes [152]. Whereas, two weeks of Zn carnosine at 70 mg·day^−1^ had no effect on 80% VO_2Peak_ performance or lactate accumulation, although this time-period was sufficient for Zn to reduce exercise-induced gut permeability by 71% (“leaky gut”; [153]). Similarly, one week of low-dose Zn gluconate at 20 mg·day^−1^ did not improve estimated VO_2peak_ (Astrand and Ryhming protocol) compared to a placebo in healthy sedentary males [154]. However, Zn supplementation reduced blood viscosity (lower viscosity is associated with aerobic exercise as it promotes oxygen delivery; [155]), which, over time, could potentially lead to improved aerobic performance. Overall, these findings show inconsistent effects of Zn supplementation on VO_2peak_.

The absence of change in endurance performance parameters may be due to the potential existence of a latent response to Zn. In elite cyclists with Zn deficiency (<11 μmol·L) receiving 22 mg·day^−1^ for 30 days and maintaining their normal training, participants’ Zn status improved and following the cessation of Zn supplementation and replacement with a placebo continued for a subsequent 30 days with plasma Zn concentrations improving further in ~1 μmol·L increments (elite cyclists; [156]). It is plausible that a longer trial period may be required to achieve more complex physiological effects with Zn. These results showed that as plasma Zn increased, Cu decreased, resulting in an increase in the Zn:Cu ratio, and this was positively correlated with an increase in insulin and HOMA2-IR (a measure of insulin resistance; 27% and 47%, respectively), indicating that Zn may impair glucose utilisation [156]. These findings could provide some rationale to the relatively inconsistent estimated VO_2peak_ results previously presented [152,153,154]. It is important to note that these data contrast the findings of improved insulin sensitivity with exercise training [157,158,159]. Zn also improves insulin sensitivity in individuals with obesity [160] and there is also evidence of high insulin sensitivity in elite athletes [161]. Therefore, caution needs to be adopted when considering these results until there is replication in elite and non-elite athletic populations.

Zn has also been shown to affect the anabolic hormone testosterone (T) in some, but not all, studies. Cinar et al. [162] demonstrated that high-dose Zn (2.5–3 mg·kg·day^−1^) appeared to increase post-exercise bound and free T (3.1 pg·dL^−1^). However, six weeks of resistance training with and without Zn induced greater changes (~5.7 pg·dL^−1^, ~4.7 pg·dL^−1^, respectively) compared to no change in the control group (no Zn or training). Contrary to the results of Cinar et al. [163], 30 days of Zn (22 mg·day^−1^; Zn gluconate) in elite Zn deficient male cyclists, during regular training, failed to change the thyroid stimulating hormone thyroxine or the active form triiodothyronine. This further supports the hypothesis that the resistance training stimuli enacted the hormonal changes observed by Cinar et al., [156] not the supplementation. Nonetheless, others have shown that Zn may contribute to exercise-induced hormonal adaptations. In a double-blind placebo controlled trial (over 4 weeks; in Zn sufficient well-trained road cyclists), Zn (30 mg; Zn sulphate) increased free T (by ~4 pg·dL^−1^) following a bout of exhaustive exercise, but did not affect resting levels [164].

Despite the popularity of Zn supplementation among athletes [31,40], there is little quality evidence that Zn can improve athletic performance and existing evidence relates to short trial periods (1–6 weeks). There is limited evidence that 20-30 mg day^−1^ may improve estimated VO_2Peak_ [152], reduce blood viscosity [154] and result in ~4 pg·dL^−1^ increase in T following high-intensity exercise [164] over a period of one to six weeks. Nonetheless, there were a number of methodological limitations in these studies, including that the VO_2peak_ was estimated and not measured. Therefore, the true impact of Zn on aerobic capacity remains unclear and requires further investigation. There are some curious results pertaining to insulin resistance in Zn deficient elite cyclists following Zn supplementation [156]. Although contrasting our current understanding of insulin during exercise [157,158,159] and the effect of Zn on insulin in other populations, this Zn-insulin paradigm in elite cyclists needs to be investigated further. Furthermore, little quality research on the effect of Zn supplementation on markers of muscle strength or hormonal responses to exercise in healthy adult populations was retrieved. Thus, in addition to investigating the effects on endurance capacity, research should focus on other markers of athletic performance, such as muscle strength and dynamic power for longer trial periods than have been investigated.

### 4.6. Sodium

Fifteen articles fulfilled the inclusion criteria for sodium (Na). These included 387 participants (females, *n* = 27), consisting of five RCT’s, none of which referred to elite athletes. Na supplementation is an effective strategy for improving exercise-induced changes in sodium balance [165,166,167,168,169,170], particularly in hot conditions [166,171,172]. However, the effectiveness of Na (in the form of loading) for enhancing athletic performance (and related physiological phenotypes) is controversial [173].

Some controlled laboratory and ‘real-world’ studies suggest that Na may be beneficial [165,166,174], whereas others show no-benefit [167,175,176,177,178] for a variety of reasons, which may include environmental, gender, dose, exercise type and training differences [165,166,179]. In a recent laboratory study (temperature approximately at 21 °C) of trained endurance athletes, capsules containing 360 mg Na and 540 mg chloride (Cl; standard table salt) were provided immediately prior to starting an exercise test (2h treadmill or cycling at 60% HR max) and at 25 minute intervals during exercise, but had no impact on time-to-exhaustion following the exercise test [175]. The largest study to investige the effect of Na on athletic performance [176] sampled South African ironman triathlon finishers in 2001 (mixed athletic level; *n* = 53 supplemented with 620 mg sodium chloride, *n* = 61 had placebo and *n* = 299 as a control group) with an air temperature ranging from 16 to 21 °C during the race and found that Na had no effect on race finishing times or other related physiological variables. However, training status was not considered (finishing time standard deviation was ~96 mins) and there are some indications that this may influence the physiological impact of Na [179]. Regardless, when comparing high- (164 mmol·L) to low-dose Na (10 mmol·L) in trained endurance athletes exercising in the heat (32 °C), high-dose Na-supplemented athletes showed a greater exercise tolerance (96.1 min vs. 75.3 min) in men and women [165,166]. This suggests that when exercising in heat, high-dose Na may improve athletic performance, although replication in studies with a placebo group is required [165,166].

Interestingly, there is some evidence from a series of studies by Zorbas and colleagues that suggest during a period of inactivity, supplementing with Na may maintain pre-inactivity VO_2peak_, even following 12 months of significantly reduced activity [169,170,180]. During a period of bed rest (from 74 km·wk^−1^ running), Na increased body mass and body fat, but surprisingly maintained VO_2peak_ (pre-rest = 66 vs. post-rest = 67 mL·kg^−1^·min^−1^), compared to a placebo [170]. Furthermore, following 364 days of reduced activity (0.7 km·day^−1^), Na maintained similar VO_2peak_ results to those of bed rest (pre-rest = 67 vs. post-rest = 68 mL·kg^−1^·min^−1^; [169]). This maintenance is proposed to result from an increased plasma volume (also shown elsewhere; [165,166]) and higher arterial blood pleasure [169,180]. This may be a consideration for injured endurance athletes to possibly maintain VO_2peak_ performance. However, given the limited data and as Na intake is linked to cardiovascular disease in the sedentary general population [181], caution and professional supervision should be considered if adopting such strategies.

Others have investigated the effects of low-dose 0.2 g·kg^−1^ Na on mean muscle power in well-trained cyclists and although not traditionally significant (*p* = 0.09), it was shown that there may be some ‘positive’ effect of Na on peak muscle power, with the Na showing ~30 watts greater than the placebo [174]. However, others have shown that arm ergonometric peak power, lactate or blood PH were not different in supplemented (0.21 g·kg^−1^ of Na) college wrestlers (*n* = 8) were compared to placebo (346 vs. 354 Watts; [178]).

Currently, there is no clear evidence that Na in healthy competitive endurance athletes, competing in temperate conditions, will have an ergogenic effect. However, there is some evidence that when exercising in high temperatures, a high-dose Na (164 mmol·L) beverage ingested prior to exercise may improve exercise tolerance [165,166]. As there is evidence that training status may significantly affect the physiological impact of Na [179], further studies similar to Huw-Butler et al. [176] with a higher dose of Na (>164 mmol·L), considering training status and assessing finishing times, would further establish the real-world impact of Na on athletic performance. Despite this, there is some interesting (although not confirmed) evidence to suggest that during a period of inactivity, supplementing with Na may maintain pre-inactivity VO_2peak_ [169,170,180]. If found to be replicable (and safe), this could be a useful strategy to maintain performance through short-term periods of injury for elite/highly-trained endurance athletes, but warrants considerable investigation regarding its safety as an addition to a rehabilitation program.

### 4.7. Selenium

Five articles fulfilled the inclusion criteria for selenium (Se). These included 124 participants (females, *n* = 0), consisting of four RCT’s, one of which referred to elite athletes. There appears to be no benefit to supplementing with additional Se on athletic performance, although research is limited.

In trained cyclists, Se (200 μg·day^−1^) over a four-week period had no additional benefit to the exercise-induced (exhaustive) increase in T or lactate accumulation [164]. Furthermore, following 10 weeks of endurance training, in previously untrained participants, consuming 180 μg of Se had no effect on mitochondrial activity, myosin heavy chain expression in muscle fibres or aerobic performance [182]. However, Se has been shown to increase glutathione peroxidase (which protects against oxidative stress) to a greater extent than a placebo, in response to exercise [183]. The absence of Se-induced exercise-adaptation to endurance exercise may not be surprising as the same group also showed that Se dampened the rate of exercise-induced mitochondrial density and overall biogenesis [184]. The authors suggested that this was potentially due to the increased cellular anti-oxidative capacity evidenced by their previous finding of increased glutathione peroxidase following exercise training [183,184]. This mechanism is logical as Se toxicity (excessive Se) can induce excessive mitochondrial oxidative stress, leading to organelle damage and dysfunction [185]. Furthermore, because mitochondrial density is a strong predictor of aerobic capacity [186], excessive Se may in fact be harmful to athletic performance, but inadequate amounts may increase exercise-induced oxidative stress over time [183,187]. However, if Se is inadequate, this may increase exercise-induced oxidative stress over time [183,187]. Savory et al. [187] investigated the effects of three weeks of sodium selenite (200 μg; supplementary form of Se) on exercise-induced (30 min at 70% VO_2peak_) oxidative stress in normal weight (NW) and overweight (OW) individuals. Se had no effect on exercise-induced markers of oxidative stress in NW, but mitigated the increase in lipid hydroperoxide (a marker of fatty acid oxidation) observed in the placebo condition in OW. It is important to note that at baseline, OW had low (according to recent population data; [188,189,190]) Se plasma concentrations (46 μg·L) compared to NW (68 μg·L) and this may suggest that individuals deficient in Se may experience greater exercise-induced oxidative stress. In this case, Se could be a potentially useful strategy to reduce chronic exercise-induced oxidative stress, considering that 11% of athletes may have deficient Se intakes [35] and that certain dietary choices that are currently growing in popularity can lead to Se deficiency, such as the Vegan diet [25].

While there appears to be no beneficial effect of supplementing with additional Se on athletic performance, current evidence suggests that it may be important to maintain an adequate Se status and that there may be an important limit [191]. Excessive Se may be harmful to athletic performance, but inadequate amounts may increase chronic exercise-induced oxidative stress [183,187]. Nonetheless, currently there are only indications as to the detrimental effects of Se deficiency on athletic performance-related phenotypes [187]. Therefore, significantly more research is needed, particularly considering the potential deficiencies that exist in athletic populations [35].

### 4.8. Chromium

Twelve articles fulfilled the inclusion criteria for chromium (Cr), including 526 participants (females, n = 249). These consisted of 11 RCT’s, none of which referred to elite athletes. Cr has been shown to be beneficial in several diseases [192], with a relatively small effect on reducing body mass in humans [193] and the majority of studies in athletic populations have focused on examining body composition-related outcomes. The most recent and well-conducted investigation (randomized, double-blind placebo controlled) supplemented a group of young (19 years) female swimmers with 400 μg of chromium picolinate over a 26-week competitive season [194]. Cr supplementation had no effect on body composition after 12/13 weeks, but following 26 weeks (training tapered during week 23–26), fat free mass (FFM) increased (~2.2 for Cr vs. 1.0 kg for placebo) and fat mass percentage deceased to a great extent compared to the placebo. In the largest study to investigate Cr and anthropometric changes in 154 adults (non-athletes) who were tracked (no training intervention) over 72 days, both 200 μg and 400 μg of Cr daily reduced percentage body fat (~1.4%), tended to increase FFM (~0.5 kg) and these results were slightly better in the 400 μg group [195]. Nonetheless, the vast majority of other data showed no association with markers of anthropometric or body composition change with [196,197,198,199,200,201,202,203] and without exercise intervention [198], in athletes [196,198,199] or non-athletes [197,198,200,201,203], in a range of doses (200–924 μg) and trial periods (6–13 weeks not including the longer studies of [194,195]).

There appears to be no benefit of Cr muscle strength and power [196,197,198,199,200,201,202,203], aerobic capacity [199], anaerobic capacity [199], fibre type [200,201] or insulin metabolism [199,204,205]. However, it should be noted that the trial periods in these studies were far shorter than studies showing modest body composition changes with Cr supplementation [194] and it may be that longer periods are required. The absence of good evidence for the benefit of Cr on human physiology has prompted some to question its efficacy as an essential mineral [206]. Overall, the current evidence of Cr supplementation for athletic performance is lacking, with most evidence showing no effect (with shorter trial periods <13 weeks), however further longer term studies may be warranted.

### 4.9. Boron

Five articles fulfilled the inclusion criteria for Boron (Br). These included 122 participants (females, *n* = 84), consisting of two RCT’s, none of which referred to elite athletes. Br is an important mineral for human health [207], however little attention has been given to the context of athletic performance. The limited evidence that exists suggests that seven weeks of Br supplementation had no effect on total-T, FFM, 1RM squat or bench press in bodybuilders [208,209] and when athletes and sedentary controls are supplemented with Br (3 mg·day^−1^ for 10 months), serum phosphorus levels were lowered [210,211]. However, this effect is diminished with exercise training [211]. There is also some evidence that Br can negatively affect Mg in athletes, but not ubiquitously [210,211].

Currently, there is no evidence to support the use of Br for any athletic performance phenotype. To the authors’ knowledge, there is no evidence in humans that Br has any potential in improving physiological athletic performance, although there is some evidence in animal models for improved bone metabolism [207], but not BMD in humans [210,212].

### 4.10. Multi Minerals

Five articles which fulfilled the inclusion criteria involved multi-minerals. These included 152 participants (females, *n* = 23), consisting of three RCT’s, one of which referred to elite athletes. The articles investigated exercise/athletic performance-related variables with combinations of minerals (in a number of biological forms) inadvertently or deliberately to investigate their cumulative effect, with mixed results [51,164,213,214,215]. 

There is some evidence of improvements in ‘real-world’ endurance performance with multi-mineral supplementation. Del Coso et al. [213] loaded 13 triathletes with an electrolyte combination of 2580 mg of Na, 3979 mg of Cl, 756 mg of K, and 132 mg of Mg, divided into three dosing intervals, during a half ironman race and compared the results to a placebo group of 13 athletes. The supplemented group had a faster cycle speed and tendency towards faster running speed along with a quicker finishing time (by ~25 min), compared to the placebo group. Others have investigated the effects of a combination of Zn-Se supplementation on blood lactate and testosterone, compared to a placebo and Zn or Se alone over four weeks in road cyclists (with 3–4 years’ experience, *n* = 32; [164]), showing no effect of Zn-Se combined on blood lactate or total and free-T pre- or post-exhaustive exercise.

Marine multi-minerals naturally enriched in calcium have also been the focus of some more recent studies. Barry et al. [214], Shea et al. [51] and Sherk et al. [215] investigated the acute effects of a mineral-rich algae compound (*Lithothamnion* species) in a number of exercise contexts. Barry et al. [214] showed that this marine-derived MTE (~12% Ca, ~1% Mg and >70 trace elements) consumed before and during a 35 km cycling time trial attenuated the exercise-induced increase in PTH in amateur athletes (*n* = 35) compared to the placebo (potentially beneficial to muscle physiology, see Ca section; [88,89]). However, there was no effect on biomarkers of bone resorption or 35 km cycling time trial performance. In a larger follow up study (*n* = 51) investigating the same supplementation and dose (1000 mg), but in a chewable form 30 minutes prior to exercise, Sherk et al. [215] found an attenuation of the exercise-induced decrease in iCa following supplementation compared to the placebo. In addition, there was a trend towards an attenuated PTH response to exercise following supplementation, but there were no effects on bone resorption. Similar results were seen in unfit postmenopausal women, where exercise (60 min of walking at 75% VO_2peak_) resulted in a decrease of iCa and an increased PTH, but this was attenuated when supplementation commenced 60 minutes before exercise (8 vs. 26 pg·mL; [51]).

There is currently little evidence for the beneficial effects of mineral combinations from either metal element combinations or natural complexes on athletic performance. Nonetheless, there is some evidence that a combination of Na, K and Mg loading may improve aspects of half ironman performance [213]. However, real-world replication is needed with complementary controlled laboratory evidence before this strategy can be recommended. Furthermore, while there is currently some evidence of attenuation of an exercise-induced increase in PTH and decrease in iCa, there is no evidence for performance effects of mineral-rich algae compounds (*Lithothamnion* species).

## 5. Limitations

The present study was conducted in accordance with the PRISMA and PICO guidelines and was written and conducted with reference to the AMSTAR 2 systematic review assessment tool [216]. Nonetheless, there are some unavoidable limitations within the present review. Due to the lack of retrieved experimental data, only Iron, Calcium, Magnesium, Phosphate, Zinc, Sodium, Selenium, Boron and Chromium MTEs were reviewed. Both controlled trials and cross-sectional studies of either athletes or non-athletic populations were included. The rationale for this decision was to incorporate as much evidence as possible of athletic performance-related phenotypes. The sample consisted of only adult non-diseased populations, which may have resulted in the elimination of a limited number of semi-relevant articles involving less serious diseases or involving younger populations. Furthermore, it was not possible to perform meta-analysis due to the wide range of outcome measures investigated in studies.

## 6. Conclusions

Currently, there is not sufficient evidence to suggest specific guidelines to assist in formulating mineral specific dietary recommendations to improve athletic performance, other than to assess baseline mineral insufficiency and ensure adherence to RDAs (while there are currently no athlete specific guidelines). In general, the scientific evidence to support the use of mineral supplementation for sports performance is lacking in volume and quality. However, there are some notable exceptions that may be better utilised under particular physiological states (deficiencies, thermoregulatory stress etc.) rather than as general ergogenic aids. Iron and Magnesium supplementation remain the minerals with the most and highest quality research, although greater advances in randomised control trials are required. Furthermore, there is a need for the replication of some key, good quality studies investigating the efficacy of particular minerals for athletic performance. In general, the conclusion and recommendations of the present review are in agreement with the International Society of Sports Nutrition [16,18], however the present review adds to the existing literature and current knowledge of MTEs and athletic performance with a systematic review, including all recent up-to-date articles of MTEs implicated to potentially have a role in athletic performance.

## 7. Key Points

Iron and Magnesium supplementation have the best quality evidence for improvements to markers and outcomes related to exercise capacity and athletic performance.In NAID females, oral supplementation of 100 mg day^−1^ ferrous sulfate or providing elemental Fe between 15–60 mg·day^−1^ over six-to-eight weeks may be adequate to elect performance adaptations in 3000 m running time, running velocity to lactate threshold, onset of blood lactate accumulation, quadriceps MVC fatigue resistance, post-fatiguing MVC, 4 km rowing time trial energy efficiency (kcal), preventing exercise-induced Fe loss, Hb, VO_2peak_, improving 15 km cycling and two mile running time trial, quicker recovery post exercise and indicators of mood. Approximately 100 mg·day^−1^ elemental Fe over 11 weeks has been shown to result in adaptations in markers of dynamic and absolute strength.300–500 mg·day^−1^ of Magnesium in the short-term (~1–4 weeks) may have a positive influence on functional dynamic measures of muscle performance (CMJ, 1RM, fatigue resistance) and longer-term (>7 weeks) benefits on quadricep torque measurements.

## Figures and Tables

**Figure 1 nutrients-11-00696-f001:**
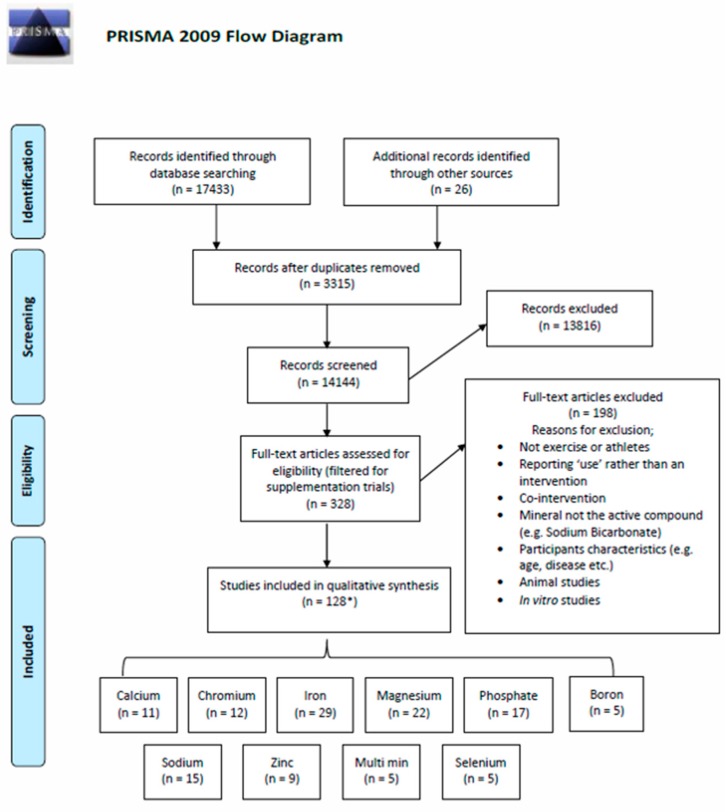
PRISMA schematic summarising the search strategy and study selection. ***** One study assessed both Zn and Se separately and in combination (*n* = 3 groups). Thus, is counted in subsections Zn, Se and multi-minerals (also accounted for in all other cumulative study sample calculations).

**Figure 2 nutrients-11-00696-f002:**
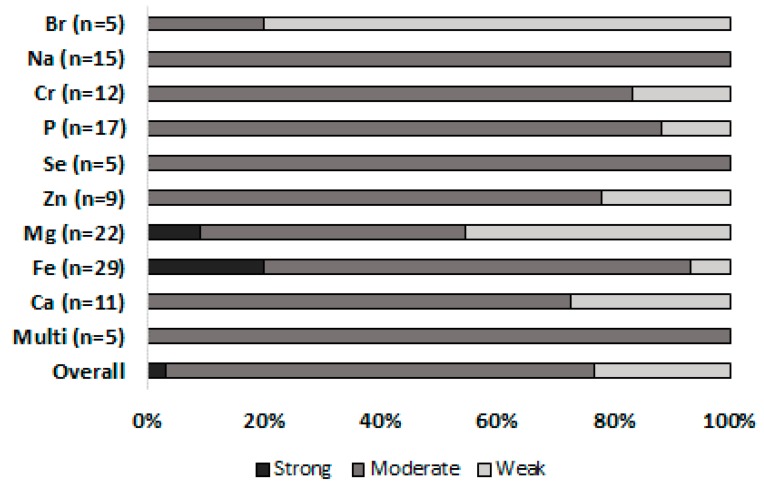
EPHPP global quality rating. Presented as percentage of articles rated as strong, moderate and weak for each mineral.

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
