# Peer review of "The Role of Mineral and Trace Element Supplementation in Exercise and Athletic Performance: A Systematic Review"

_nutrients, 2019, doi:10.3390/nu11030696_

Round 1
Reviewer 1 Report
The aim of this article was to review the role of all mineral and trace elements (MTEs) on exercise and physical performance. It has achieved this. The strengths of this paper are that it reviews all of the research that looks at MTEs and physical activity and collates them in one paper. The novel aspect of this paper is that this is the first time that all MTEs have been put together in one paper. However, I feel that the introduction needs to define the different types of performance as to just say 'Exercise and Athletic Performance' is very broad. I would suggest that a short paragraph is added in the introduction to discuss different types of exercise and within each micronutrient it would be helpful to say what type of performance would be helped by the specific MTE.The paper is very large and contains a lot of information. The latter MTEs do not show much in effect on performance and exercise. It would be helpful to the reader if the summary could be more specific about the MTEs that do help performance and how they help it, just the key points so that the reader can get this information without having to read all of the information.
Overall this is a good systematic review and will add to the literature. I have made some specific points below.
page 2 . First line doesn't make sense. If not adhering to RDAs does not strictly result in deficiencies why would certain dietary choices lead to deficiencies?
intro 3rd paragraph. 'some athletes......magnesium' this is a random sentence and does not fit in with the paragraphs around it
3rd paragraph 4th line ' similar to the available population data' you have not mentioned this yet, so the reader does not know it is similar.
4th paragr4aph ' However, there may be ergogenic properties of MTEs in achieving or possibly surpassing the RDA that are designed for the general population' I'm not sure what you are saying here.
In the PRISMA flow chart you have said that additional records were identified through other sources. I can't find what these other sources are.
Author Response
Open Review(reviewer one)
Reviewer comment
The aim of this article was to review the role of all mineral and trace elements (MTEs) on exercise and physical performance. It has achieved this. The strengths of this paper are that it reviews all of the research that looks at MTEs and physical activity and collates them in one paper. The novel aspect of this paper is that this is the first time that all MTEs have been put together in one paper. However, I feel that the introduction needs to define the different types of performance as to just say 'Exercise and Athletic Performance' is very broad. I would suggest that a short paragraph is added in the introduction to discuss different types of exercise and within each micronutrient it would be helpful to say what type of performance would be helped by the specific MTE.The paper is very large and contains a lot of information. The latter MTEs do not show much in effect on performance and exercise. It would be helpful to the reader if the summary could be more specific about the MTEs that do help performance and how they help it, just the key points so that the reader can get this information without having to read all of the information.
Authors response
Thank you for your comments and suggestions. We have now included additional information on different types of exercise in the introduction that now reads;
“Specifically, the effects of MTEs on those phenotypes that benefit general markers of performance, for example lower body power output, maximal endurance capacity, maximal and relative muscle mass and strength, and fatigue recovery capacity.”
Also, we have added an mini conclusion (suggested by reviewer 2) to each MTE section that includes the type of performance would be helped by the specific MTE and the key points for the reader. For example the text for Magnesium reads;
“Overall, the current evidence suggests that short-term supplementation (~1-4 weeks) can have a positive influence on functional dynamic measures of muscle performance (CMJ, 1RM)…”
Additionally, we have included a “Key points” section just after the conclusion, this reads;
“Key points
Iron and Magnesium supplementation have the best evidence for improvements to exercise capacity and athletic performance.
For non Iron deplete individuals, >325 mg·d-1of oral Iron can elect beneficial physiological adaptation to endurance performance and in Iron-deplete, 160 mg·d-1may improve aerobic capacity.
300-500 mg·d-1of Magnesium over short-term (~1-4 weeks) can have a positive influence on functional dynamic measures of muscle performance (CMJ, 1RM) and over longer term (>7 weeks) could elect additional neuromuscular adaptation.”
Reviewer comment
Overall this is a good systematic review and will add to the literature. I have made some specific points below.
Reviewer comment
page 2 . First line doesn't make sense. If not adhering to RDAs does not strictly result in deficiencies why would certain dietary choices lead to deficiencies?
Authors response
Thank you for your comment. We were differentiating between RDA intake and biological deficiencies however to clarify this point we have amended the text that now reads;
“While not adhering to RDAs does not strictly result in biological deficiencies, certain dietary and lifestyle choices may introduce additional challenges to RDA adherence and lead to deficiencies.”
Reviewer comment
intro 3rd paragraph. 'some athletes......magnesium' this is a random sentence and does not fit in with the paragraphs around it
Authors response
Thank you for your comment. We have amended the text to link better with the text around it, that now reads;
“For example, some athletes (n=25 male Polish) may…”
Reviewer comment
3rd paragraph 4th line ' similar to the available population data' you have not mentioned this yet, so the reader does not know it is similar.
Authors response
Thank you for your comment. We have amended the text to include some specific population data and now reads;
“Similar to the available population data [for example, 15%deficiency ofMg; 22-25]…”
Reviewer comment
4th paragr4aph ' However, there may be ergogenic properties of MTEs in achieving or possibly surpassing the RDA that are designed for the general population' I'm not sure what you are saying here.
Authors response
Thank you for your comment. We are suggesting that the current RDAs, that are designed for general population health may not be appropriate for athletes who regularly participate in high intensity physical activity (i.e. may require more micronutrients on account f greater metabolic usage). We have amended the text to make this point more clear which now reads;
“However, there may be ergogenic properties of MTEs in achieving or possibly surpassing the RDA, that are specifically designed for the general population health.”
Reviewer comment
In the PRISMA flow chart you have said that additional records were identified through other sources. I can't find what these other sources are.
Authors response
Thank you for your comment. The other sources were articles that the authors were aware of prior to undertaking the review (that were not retrieved by the review), hand searched reference lists and other reviews.

Reviewer 2 Report
Authors present a systematic review in relation to minerals and trace elements and their role with physical performance.
The truth is that it is an extensive revision that could be given to write a book because of the large volume of data presented by the authors. However, I believe that this large amount of information makes it too widespread. I would propose to the authors to do meta-analysis reviews of each mineral separately. Thus, tables would be presented in which it would be shown on which elements of sports performance have shown improvement; in addition to showing conclusions individually giving information on effective doses, as well as periods of treatment.
If the authors do not see this fact feasible, I believe that the review should be oriented to a narrative review on the subject, since relevant information of each nutrient is lost especially in matters related to its metabolism, which could facilitate reaching a more specific conclusion . If I would propose that in each nutrient a mini conclusion be given, beyond the general one at the end of the article. Likewise, the authors must correctly specify the dates on which the revision is made, since it seems that it has only been for a few months of 2018.
Author Response
Open Review(reviewer Two)
Reviewer comment
Authors present a systematic review in relation to minerals and trace elements and their role with physical performance.
The truth is that it is an extensive revision that could be given to write a book because of the large volume of data presented by the authors. However, I believe that this large amount of information makes it too widespread. I would propose to the authors to do meta-analysis reviews of each mineral separately. Thus, tables would be presented in which it would be shown on which elements of sports performance have shown improvement; in addition to showing conclusions individually giving information on effective doses, as well as periods of treatment.
Authors response
Thank you for your comment and suggestion. However, respectfully, a meta-analysis is not feasible nor was it the intended purpose or scope of the present review which was to systematically review and critically evaluate all available literature, give the inclusion criteria. The authors discussed the option of a meta-analysis and determined that it was not possible, particularly due to the diversity and lack of replication of particular athletic performance traits for individual minerals, this is evident in the manuscript text (see main text and Supplementary table 1). Nonetheless, within the review, information about dose and treatment period are discussed for each mineral where it is appropriate and is presented, for example;
“Nonetheless, to elicit changes in endurance related performance outcomes, greater than 3 days of loading with ~4 g·day-1is required and endurance performancecan be further maintained with a lower dose [141,142].”
Also, as requested by reviewer one we have included an additional section after the conclusion “Key Points” that reaffirms the important findings of the present review, including dose and supplementation time (see revised manuscript).
However, given the reviewers comments, we have amended the text to elaborate on these points and include more details on dose and supplementation time, where appropriate (see revised manuscript).
Reviewer comment
If the authors do not see this fact feasible, I believe that the review should be oriented to a narrative review on the subject, since relevant information of each nutrient is lost especially in matters related to its metabolism, which could facilitate reaching a more specific conclusion. If I would propose that in each nutrient a mini conclusion be given, beyond the general one at the end of the article. Likewise, the authors must correctly specify the dates on which the revision is made, since it seems that it has only been for a few months of 2018.
Authors response
Thank you for your comments and suggestions. Unfortunately, the reviewer may have misinterpreted the aim and the main target audience for the present review. In this light, we have re-examine and edit the introduction to insure that other readers, specifically those interested in the role that MTEs might play on their (and/or individuals they advise) ‘practical’ athletic performance, do not make the same interpretation (see track changes in introduction for amendments). The present review was performed systematically in accordance with the PICO and PRISMA guidelines to specifically assess “the role of minerals and trace elements on athletic performance”, thus the retrieved articles were not were not selected narratively and were reviewed specifically with reference to systematic review guidelines. Furthermore, given the aim of the present article, the authors (as does reviewer 1) believes that the adopted approach reaches justified and specific conclusions relevant to the aim.
The authors have specifically chosen the present aim as it is the functional outcomes i.e. those that are reflected in the specific ‘functional’ role (e.g. benefit) that an MTE may have on markers of athletic performance (not their potential metabolic pathways that may or may not be relevant). It is this information that is of importance to individuals that are considering investing time and capital on MTEs and the absence of this knowledge or access to the information is at least partly remedied in the synthesized (open access – if accepted in Nutrients) information of the present review.
Is possible that the absence of this knowledge is the reason that >50% of athletes currently consume unregulated supplements (see introduction in the present review) that, for the most part, have no practical function on athletic performance or related phenotypes (see discussion in the present review) regardless of potential metabolic affects. We specifically elude to this in the introduction. (kindly see the article synopsis from reviewer one where it is stated that the authors have achieved this specific aim and the reviewer discuss the strengths of the present review in relation the this aim and collation of multiple MTEs in a single article). Discussing the details of potential metabolic effects of each individual mineral, would not have facilitated the aim of the present review – this type of review would require an alternative aim (or set of aims) and is a worthwhile topic but not the purpose or scope of the present article.
These things being said, the reviewer highlights that a “mini conclusion” for each mineral would be useful to the reader. While this is present for each mineral throughout the text, the authors included additional text to facilitate more specific conclusions, that are relevant to the aim of the present review and where appropriate. The review dates are actuate and appropriately referenced within the text.

Round 2
Reviewer 2 Report
The authors have made a great effort to resolve the doubts. However, I consider important a mini conclusion in each of the trace elements commented so that the doses, duration, timing and effects on atheltic performance are specified.
Author Response
Reviewer comment
The authors have made a great effort to resolve the doubts. However, I consider important a mini conclusion in each of the trace elements commented so that the doses, duration, timing and effects on athletic performance are specified.
Authors response
Thank you for your further comments and acknowledgement of the authors efforts so far. We now agree fully with the reviewer and have included a mini conclusion in each of the trace elements so that the doses, duration, timing and effects on athletic performance are specified. However as there was no evidence that Selenium, Chromium, boron can effect athletic performance we could not include information about doses, duration, timing. We hope these amendments facilitate the reviewers comments, which now read as fellows;
Iron
In conclusion, there appears to be a range of factors including baseline iron status, dose and rout of administration that may influence the efficacy of iron supplementation on iron status and performance. It is generally accepted that Fe increases HBmass leading to greater oxygen delivery, however there may be other mechanisms not related to erythropoiesis and oxygen transport at play [i.e. no change in HBmass in non-Fe deficient athletes; 77,79]that over longer trial periods could lead to performance enhancement. The general trend evident as is also evident in recent systematic reviews is that Fe supplementation may benefit iron status and athletic performance in individuals with compromised iron status [42,43]. In NAID females, oral elemental Fe supplementation between ~15-60 mg·d-1or 100mg.d-1 of ferrous sulfate over 6-8 weeks may be adequate to elect performance adaptations in; 3000m running time, running velocity to lactate threshold, onset of blood lactate accumulation, quadriceps MVC fatigue resistance, post-fatiguing MVC, 4 km rowing time trial energy efficiency (kcal), prevent exercise-induced Fe loss, Hb, VO2peak, improve 15km cycling and two mile running time trial, quicker recovery post exercise and indicators of mood. Approximately 100 mg·d-1 elemental Fe over 11 weeks has been shown to result in adaptations in markers of dynamic and absolute strength. Intravenous Fe administration may be beneficial to improving running time and VO2max performance, however the current evidence on whether Fe IV infusion or injections are superior to oral supplementation is limited and the administration (i.e. via a medical professional) makes it difficult to recommend at this time. Future studies should focus on recruiting larger samples, including elite athletes, tracking longer term supplementation and considering alternative mechanisms to the potential changes in HBmass. As the majority of individuals supplementing with Fe are not elite athletes, investigations of Fe-associated training adaptations in non-elite individuals are also currently limited and warrant attention.
Calcium
Although Ca is vitally important for muscle and cardiovascular function [98,99]there is currently no evidence that Ca supplementation has any direct effect on athletic performance (currently, only aerobic capacity has been investigated). Nonetheless, calcium supplementation at oral doses between 800 (over 8 days)-1352mg (single meal prior to exercise) , or IV infusion at 156mg (prior to and during exercise) may attenuate post-exercise reductions in serum iCa and Ca loss, with lower doses appearing to have no effect. Supplementary Ca may also reduce the exercise-induced increase in iPTH. As mentioned above, the impact on PTH could potentially have implications for muscle strength [88,89], however there is currently no direct evidence to support the hypothesis that Ca supplementation may enhance muscle physiological capacity though the actions of PTH. Future research should investigate this using detailed measures such as hormonal changes, muscle fibre characteristics, muscle-derived biochemical effects, action potentials, functional and specific strength measures.
Magnesium
In conclusion, the current evidence suggests that 300-500 mg·d-1 for short-term supplementation (~1-4 weeks) can have a positive influence on functional dynamic measures of muscle performance (CMJ, 1RM and fatigue resistance) and exercised-induced inflammation, DNA damage, cortisol and immunological blood markers, but no effect on isokinetic performance [107,116], 118. Whereas, longer supplementation trials (~7 weeks) can elicit training-induced adaptive responses in young untrained populations [103] – whether this would be reflected in well-trained athletic populations is yet to be established. Furthermore, Mg (300 mg·d-1 over 12 weeks) may improve markers of functional performance in older populations and may be a consideration to maintain functional capacity throughout aging, but may require an even longer treatment period [127]. Lastly, there is currently no evidence to support a benefit of Mg supplementation to improve endurance performance related outcomes, despite a logical biological potential [115]. However, the literature is limited and further research is required. Mg appears to have some ergogenic potential but much more evidence is needed in a variety of populations (untrained, elite athletes and elderly) and in response to both aerobic and resistance/dynamic power training and performance. Little is currently known about the direct effect of Mg ingestion on muscle in response to exercise and further investigation is needed to uncover the mechanisms of action for the above physiological responses.
Phosphate
In conclusion, the current evidence indicates several ergogenic effects of SP supplementation with ~4 g·day-1 over 3-6 days on a range of performance related outcomes such as sprint time, cycling power output, VO2Peak, resting HR, biomarkers of metabolic demand and measures of cardiac function (echocardiographic). However, these ergogenic effects may be limited to highly-trained individuals and may not aid recreational athletes when dietary phosphorus is adequate and outside of the requirements of highly-trained athletes. In addition, benefits in endurance performance related outcomes can be further maintained with a lower dose of ~2 4 g·day-1 [141,142]. As there is some evidence for a delayed response to the adaptation of muscle power [142], this is an interesting area for future investigation, along with determining if this is a central or peripheral physiological adaptation and if it is directly related to changes in 2,3-DPG. Furthermore, Kreider et al. [135] presented some possible mechanistic data in relation to morphological changes to the myocardium in a small sample, this requires replication with a larger sample in future investigations.
Zinc
Despite the popularity of Zn supplementation among athletes [31,40] there is little quality evidence that Zn can improve athletic performance and existing evidence relates to short trial periods (1-6 weeks). There is limited evidence that 20mg.d-1may improve estimated VO2Peak [152], reduce blood viscosity (ref 154 here),and 30 mg.d-1increase T following high intensity exercise (ref 164) over a period of 1-6 weeks. Nonetheless, there were a number of methodological limitations in these studies including that VO2peak was estimated and not measured. Therefore, the true impact of Zn on aerobic capacity remains unclear and requires further investigation. There are some curious results pertaining to insulin resistance in Zn deficient elite cyclists following Zn supplementation [156]. Although contrasting our current understanding of insulin during exercise [157-159] and the effect of Zn on insulin in other populations, this Zn-insulin paradigm in elite cyclists needs to be investigated further. Furthermore, little quality research on the effect of Zn supplementation on markers of muscle strength or hormonal responses to exercise in healthy adult populations was retrieved. Thus, in addition to investigating effects on endurance capacity, research should focus on other markers of athletic performance such as muscle strength and dynamic power for longer trial periods than have been investigated.
Sodium
Currently there is no clear evidence that Na in healthy competitive endurance athletes, competing in temperate conditions will have an ergogenic effect. However, there is some evidence that when exercising in high temperatures, a high dose Na (164 mmol·L) beverage ingested prior to exercise may improve exercise tolerance [165,166]. As there is evidence that training status my significantly affect the physiological impact of Na [179], further studies similar to Huw-Butler et al. [176] with a higher dose of Na (>164 mmol·L), considering training status and assessing finishing times would further establish the real-world impact of Na on athletic performance. Despite this, there is some interesting (although not confirmed) evidence to suggest that during a period of inactivity, supplementing with Na may maintain pre-inactivity VO2peak [169,170,180]. If found to be replicable (and safe) this could be a useful strategy to maintain performance through short term period of injury for elite\highly-trained endurance athletes, but this warrants considerable investigation regarding its safety as an addition to a rehabilitation program.
Key points
• Iron and Magnesium supplementation have the best quality evidence for improvements to markers and outcomes related to exercise capacity and athletic performance.
• In NAID females, oral supplementation of 100mg.d-1 of ferrous sulfate or providing elemental Fe between 15-60 mg·d-1over 6-8 weeks may be adequate to elect performance adaptations in; 3000m running time, running velocity to lactate threshold, onset of blood lactate accumulation, quadriceps MVC fatigue resistance, post-fatiguing MVC, 4 km rowing time trial energy efficiency (kcal), prevent exercise-induced Fe loss, Hb, VO2peak, improve 15km cycling and two mile running time trial, quicker recovery post exercise and indicators of mood. Approximately 100 mg·d-1 elemental Fe over 11 weeks has been shown to result in adaptations in markers of dynamic and absolute strength
• 300-500 mg·d-1of Magnesium over short-term (~1-4 weeks) may have a positive influence on functional dynamic measures of muscle performance (CMJ, 1RM, fatigue resistance) and over longer term (>7 weeks) benefits on quadriceps torque measurements.